# An efficient class of estimators for finite population mean in the presence of non-response under ranked set sampling (RSS)

**Syed Abdul Rehman** [1]*, **Javid Shabbir** [2,3]

**1** Department of Mathematical Sciences, Balochistan University of Information Technology, Engineering and Management Sciences, Quetta, Pakistan, **2** Department of Statistics, Quaid-i-Azam University, Islamabad, Pakistan, **3** Department of Statistics, University of Wah, Rawalpindi, Pakistan

These authors contributed equally to this work.
* rehmanbukhari725@gmail.com

**Data Availability Statement:** All relevant data are within the paper.

**Funding:** The authors received no specific funding for this work.

## Abstract

In this study, we address the problem of estimating the finite population mean when the non-response occurs on the characteristics under study. We propose a class of Rao-regression type estimators when ranked set sampling (RSS) procedure is used to collect the data from non-response group only and from both, the response and non-response groups. The information provided on the auxiliary variable is used at both stages i.e., at designing stage and the estimation stage. Expressions for bias and mean square error of the estimators are obtained up to first order of approximation. A comprehensive simulation study is carried out to observe the performances of the estimators under non-response.

## Introduction

A survey can be conducted by using the variety of sampling techniques depending upon situation and structure of the data. Simple random sampling (SRS) is an easy and unbiased data collection method, but in situations where the overall population is scattered and diverse, it requires a larger sample for a given margin of accuracy. On the other hand, estimation of population parameters becomes more problematic if non-response occurs in a scattered population. Non-response is an inability to collect data from one or more units of a population selected in a sample frame. Most of the time, we have to generalize the survey results by using a smaller sample, which demands that an appropriate sampling method should be employed to collect the sample. In such case, researchers tend to employ robust methods for survey design and parameter estimation. For example, some writers have proposed estimators of the population mean based on L-moments theory, which is robust to outliers and less vulnerable to the effects of sample volatility. The L-moments theory includes the use of order statistics, and it leads to new procedures for estimating population parameters. Computation of the first few sample L-moments and their ratios provides a useful review of the location, shape, and dispersion of the population from which the sample was drawn. In this regard, the work of [1–4] can be seen. Another efficient and alternative approach of estimating population parameters is

**Competing interests:** The authors have declared that no competing interests exist.

ranked set sampling (RSS) procedure, which is determined by ranking a greater number of sampling units based on their relative sizes, then picking a smaller number of units from each ranked group under observation. As a result, RSS increases the precision of estimates by reducing sampling error. The availability of some appropriate auxiliary variables that are correlated with the variable under study is also a factor in improving the estimation. The RSS sampling procedure utilizes the auxiliary information to collect data by using its order at the designing stage.

Theory of RSS was first introduced by [5]. In this method, units of the study variable are ordered by either visual judgment or by some cheap quantitative measures. [6] developed an unbiased estimator of the population mean under RSS technique. [7] discovered another use of the auxiliary variable, and it is that one can use its order to rank units of the study variable. [8–18] can also be seen for the use of this feature of an auxiliary variable. Use of the auxiliary variable can also be used to reduce sample variation and increase the efficiency of estimates at the estimation stage. Several authors have claimed that use of the auxiliary variable at the designing stage can result in more efficient estimates. For the applications of RSS in various disciplines of research, one can review the work of [19–23].

[24] has considered the comprehensive layout of the non-response under RSS. In his book, the problem of non-response when the sample is collected through RSS on the second attempt from a non-response group can be seen. Inspired by [24–27], we investigate the estimation of finite population mean under RSS in the presence of non-response.

## Importance of the problem

Non-response is thought to be one of the most important problems in the theory of survey sampling. The occurrence of non-response leads to skewed estimates and a less representative sample of the population. The discrepancy between respondents and non-respondents on a given measure, combined with the non-response rate in the population, produces non-response bias. A lower response rate raises the risk of larger non-response bias, but when data is missing at random (MAR), a lower response rate has no effect on the non-response error. In practice, using information from the auxiliary variable is often costly, but classifying things based on it, is quite straightforward. We assume that the RSS technique may improve the precision in estimating the population mean by using information from the auxiliary variable at the estimation stage.

## Methodology

We consider the naive model of [28]. According to which, we draw a sample $S$ of size $n_s$ by using SRSWOR method from a finite population $\Omega$ of size $N$. Let $n_1$ units respond to the survey at first attempt while $n_2 (= n - n_1)$ units do not respond. Special efforts are made to approach the non-responding units and a part of them $(n'_2 = n_2/k; k > 1)$ is included in the sample. Thus, we get a final sample of size $n = n_1 + n'_2$ for estimation purpose. This allows the entire population to be divided into two complementary groups, called response and non-response groups.

Let $\left(Y_{ji}, X_{ji}\right)_{i=1}^{N_j}; j = 1, 2$ be population units of the study variable $(Y)$ and the auxiliary variable $(X)$ in the two groups. Our goal is to estimate the finite population mean $(\bar{Y})$ of the study variable. [28] suggested the following unbiased estimator when non-response occurs on $Y$:

$$\bar{y}^*_{srs} = w_1 \bar{y}_1 + w_2 \bar{y}'_2. \tag{1}$$

The variance of $\bar{y}_{srs}^*$ is given by

$$Var\left(\bar{y}_{srs}^*\right) = \left(\frac{1-\lambda}{n}\right)\sigma_y^2 + \frac{W_2(k-1)}{n}\sigma_{y2}^2 = V_{ys}, \tag{2}$$

where

$$\bar{y}_1 = \frac{1}{n_1}\sum_{i=1}^{n_1} y_{1i}, \bar{y}_2' = \frac{1}{n_2'}\sum_{i=1}^{n_2'} y_{2i}', \lambda = \frac{n}{N}, w_j = \frac{n_j}{n}; j = 1, 2$$

and

$$\bar{Y}_1 = \frac{1}{N_1}\sum_{i=1}^{N_1} Y_{1i}, \bar{Y}_2 = \frac{1}{N_2}\sum_{i=1}^{N_2} Y_{2i}, \bar{Y} = \frac{1}{N}\sum_{i=1}^{N} Y_i = W_1\bar{Y}_1 + W_2\bar{Y}_2, \sigma_y^2 = \frac{1}{N-1}\sum_{i=1}^{N} (Y_i - \bar{Y})^2,$$

$$\sigma_{y2}^2 = \frac{1}{N_2 - 1}\sum_{i=1}^{N} (Y_{2i} - \bar{Y}_2)^2, W_j = \frac{N_j}{N}.$$

Similar results can be obtained for an auxiliary variable $X$ when it is involved in the estimation of population mean for the study variable $Y$ with the following covariance:

$$Cov\left(\bar{y}_{srs}^*, \bar{x}_{srs}^*\right) = \frac{1-\lambda}{n}S_{yx} - \frac{W_2(k-1)}{n}S_{yx2}^2 = V_{yxs}.$$

When population mean of the auxiliary variable is known and incomplete information exists only on the study variable, then the [29, 30] suggested the following ratio and regression estimators for estimating $\bar{Y}$ as

$$\bar{y}_{ratio} = \bar{y}^*\left(\frac{\bar{X}}{\bar{x}}\right), \tag{3}$$

and

$$\bar{y}_{reg} = \bar{y}^* + \hat{\beta}_{yx}(\bar{X} - \bar{x}). \tag{4}$$

[31] defined the following ratio and regression estimators when non-response occurs on both variables:

$$\bar{y}_{ratio}^* = \bar{y}^*\left(\frac{\bar{X}}{\bar{x}^*}\right), \tag{5}$$

and

$$\bar{y}_{reg}^* = \bar{y}^* + \hat{\beta}_{yx}^*(\bar{X} - \bar{x}^*), \tag{6}$$

where $\hat{\beta}_{yx} = \frac{s_{yx}}{s_x^2}$ and $\hat{\beta}_{yx}^* = \frac{s_{yx}^*}{s_x^{*2}}$ are estimates of population regression coefficient $\beta_{yx} = \frac{S_{yx}}{S_x^2}$ such

that

$$
s_{yx} = \frac{\left(\sum_{i=1}^{n_1} y_{1i} x_{1i} + k \sum_{i=1}^{n'_2} y'_{2i} x'_{2i} - n \bar{y}^* \bar{x}\right)}{n-1}, \quad s_x^2 = \frac{\left(\sum_{i=1}^{n_1} x_{1i}^2 + k \sum_{i=1}^{n'_2} x_{2i}'^2 - n \bar{x}^2\right)}{n-1},
$$

$$
s_{yx}^* = \frac{\left(\sum_{i=1}^{n_1} y_{1i} x_{1i} + k \sum_{i=1}^{n'_2} y'_{2i} x'_{2i} - n \bar{y}^* \bar{x}^*\right)}{n-1}, \quad s_x^{*2} = \frac{\left(\sum_{i=1}^{n_1} x_{1i}^2 + k \sum_{i=1}^{n'_2} x_{2i}'^2 - n \bar{x}^{*2}\right)}{n-1},
$$

where

$$
\bar{x}^* = w_1 \bar{x}_1 + w_2 \bar{x}'_2. \tag{7}
$$

[27] proposed the following Rao-regression type estimators in the line of [32].

$$
t_{s1} = d_1 \bar{y}^* + d_2(\bar{X} - \bar{x}^*), \tag{8}
$$

where $d_1$ and $d_2$ are scalars, whose values are either pre-determined or calculated wisely to minimize the MSE of the estimator. The minimum MSE of $t_{s1}$ with optimum values of $d_1$ and $d_2$, is given by

$$
d_{1opt} = \frac{\bar{Y}^2 V_{xs}}{\bar{Y}^2 V_{xs} + V_{xs} V_{ys} - \rho_{yx} V_{yx} V_{xs}}, \quad d_{2opt} = \frac{\bar{Y}^2 \rho_{yx} \sqrt{V_{yx} V_{xs}}}{\bar{Y}^2 V_{xs} + V_{xs} V_{ys} - \rho_{yx} V_{yx} V_{xs}}
$$

and

$$
MSE(t_{s1})_{\min} \cong \frac{\bar{Y}^2(V_{xs} V_{ys} - \rho_{yx} V_{yx})}{\bar{Y}^2 V_{xs} + V_{xs} V_{ys} - \rho_{yx} V_{yx}}. \tag{9}
$$

[27] also proposed the following generalized class of Rao-regression type estimators by using the idea of [26]:

$$
t_{si} = (d_3 \bar{y}^* + d_4 \bar{u})h(\bar{u}), \tag{10}
$$

where $d_3$ and $d_4$ are constants and $h$ is generic function of $\bar{u} = \bar{X} - \bar{x}^*$ satisfying some mild conditions. The optimum values of $d_3$ and $d_4$ along with the minimum MSE, are given by

$$
d_{3opt} = \frac{\bar{Y}^2 V_{xs}(a^2 - 2b^2 V_{xs} + ac V_{xs})}{a[a^2(\bar{Y}^2 V_{xs} + V_{xs} V_{ys} - V_{yxs}^2) + \bar{Y}^2(2ac - 3b^2)V_{xs}^2]},
$$

$$
d_{4opt} = \frac{\bar{Y}[a^3 \bar{Y} V_{yxs} + G - 2ab^2 \bar{Y} V_{xs} V_{yxs} + b^3 \bar{Y}^2 V_{xs}^2]}{a^2[a^2(\bar{Y}^2 V_{xs} + V_{xs} V_{ys} - V_{yxs}^2) + \bar{Y}^2(2ac - 3b^2)V_{xs}^2]}
$$

and

$$
MSE(t_{si})_{\min} = \frac{\bar{Y}^2[a^4 V_{ys}^2(V_{ys} V_{xs} - V_{yxs}^2) - H + 2ab^2 c \bar{Y}^2 V_{xs}^4 - b^4 \bar{Y}^2 V_{xs}^4]}{a^2[a^2(\bar{Y}^2 V_{xs} + V_{xs} V_{ys} - V_{yxs}^2) + \bar{Y}^2 V_{xs}^2(2ac - 3b^2)]}, \tag{11}
$$

where

$$
G = a^2\{c\bar{Y} V_{xs} V_{yxs} - b(\bar{Y}^2 V_{xs} - V_{ys} V_{xs} + V_{yxs}^2)\}
$$

and

$$H = a^2 V_{xs} \{ b^2 (V_{ys} V_{xs} - V_{yxs}^2) + c^2 \bar{Y}^2 V_{xs}^2 \}.$$

[27] showed that this class of estimators is more precise than usual [28], ratio and regression estimator as discussed above under when some certain conditions are satisfied. It should be noted that the values of $a$, $b$ and $c$ are $\{1, -1, 1\}$ and $\{1, -\frac{1}{2}, \frac{3}{8}\}$ for $h(\bar{u}) = \left(\frac{\bar{X}}{\bar{X} - \bar{u}}\right)$ and $h(\bar{u}) = \log\left(\frac{\bar{X}}{2\bar{X} - \bar{u}}\right)$ respectively.

For more detail see [26, 27].

## RSS procedue

The RSS procedure is described as in the following steps:

**Step 1**: Collect $v$ independent sets each of $v$ units.

**Step 2**: Array each set inside in ascending order by mean of the study variable or any closely related auxiliary variable. The ranking is done either by visual inspection or some quantitative measurements.

**Step 3**: Select the lowest order unit from the first set.

**Step 4**: Select the second lowest order unit from the second set and continue selecting units in this way until $v^{th}$ order statistic is selected from $v^{th}$ set.

This completes one cycle of an RSS procedure which can be repeated $r$ times to obtain a sample of size $n = rv$. The selected units can be represented as $Y_{1(1)j}$, $Y_{2(2)j}$, . . ., $Y_{i(i)j}$, . . ., $Y_{v(v)j}$; $i = 1, 2, \ldots, v, j = 1, 2, \ldots, r$.

## RSS under non-response

[24] has suggested a very comprehensive work on dealing with the missing observations under RSS. Here we discuss the following two major situations of dealing non-response under RSS.

**Situation-I**: When RSS is used at second attempt only.

Let we collect data at second attempt by the method of RSS in such a way that $v$ independent sets, each of size $v$ are selected from the non-response group. The later procedure is followed step by step as discussed in Section 3 earlier. The estimate of sample mean from non-responding units based on $n'_2 = r'_2 v$ sampled units under RSS, is given by

$$\bar{y}'_{2rss} = \frac{1}{n'_2} \sum_{j=1}^{r'_2} \sum_{i=1}^{v} y'_{2(i)}. \tag{12}$$

If the population is symmetrically distributed then $\bar{y}'_{2rss}$ is an unbiased estimator of the population mean with the following expected variance

$$Var\left(\bar{y}'_{2rss}\right) = \frac{\sigma_{2y}^2}{r'_2 v} - \frac{1}{r'_2 v^2} \sum_{i=1}^{v} \Delta_{2y(i)}^2, \tag{13}$$

where

$$\Delta_{2y(i)} = \mu_{2y(i)} - \mu_{2y}.$$

Thus, Eq (1) becomes

$$\bar{y}^*_{rss} = w_1 \bar{y}_1 + w_2 \bar{y}'_{2rss}. \tag{14}$$

The estimator $\bar{y}_{rss}^{*}$ is unbiased for which the variance is given by

$$Var\left(\bar{y}_{rss}^{*}\right) = \frac{\sigma_y^2}{n} + \frac{W_2(k-1)}{n}\sigma_{2y}^2 - W_2k\left(\frac{1}{nv}\sum_{i=1}^{v}\Delta_{2y[i]}^2\right) = V_{yr}'. \tag{15}$$

Similar result can be obtained for the auxiliary variable $X$ as

$$Var\left(\bar{x}_{rss}^{*}\right) = \frac{\sigma_x^2}{n} + \frac{W_2(k-1)}{n}\sigma_{2x}^2 - W_2k\left(\frac{1}{nv}\sum_{i=1}^{v}\Delta_{2x(i)}^2\right) = V_{xr}'. \tag{16}$$

**Situation-II**: When RSS is used at both attempts.

It is an extension of Situation-I in which we collect a sample from both groups by using RSS.

The estimator of sample mean can be written as

$$\bar{y}_{rss}^{**} = w_1\bar{y}_{1rss} + w_2\bar{y}_{2rss}', \tag{17}$$

where $\bar{y}_{1rss}$ is the sample mean based on $n_1$ units collected at first attempt, while $\bar{y}_{2rss}'$ is the sample mean obtained from $n_2'$ units collected at second attempt. $\bar{y}_{rss}^{**}$ is also an unbiased estimator, the variance of $\bar{y}_{rss}^{**}$ given by

$$Var\left(\bar{y}_{rss}^{**}\right) = \frac{\sigma_y^2}{n} - \frac{1}{rv^2}\sum_{i=1}^{v}\Delta_{y[i]}^2 + \frac{W_2(k-1)}{rv}\left(\sigma_{2y}^2 - \frac{1}{v}\sum_{i=1}^{v}\Delta_{2y[i]}^2\right) = V_{yr}''. \tag{18}$$

Similar expressions can be obtained for the auxiliary variable as

$$Var\left(\bar{x}_{rss}^{**}\right) = \frac{\sigma_x^2}{n} - \frac{1}{rv^2}\sum_{i=1}^{v}\Delta_{x(i)}^2 + \frac{W_2(k-1)}{rv}\left(\sigma_{2x}^2 - \frac{1}{v}\sum_{i=1}^{v}\Delta_{2x(i)}^2\right) = V_{xr}''. \tag{19}$$

Note that the set size $v$ is kept constant while other notations are used as
$n_1 = r_1v, \quad n_2 = r_2v, \quad n_2' = r_2'v, \quad r = r_1 + r_2', \quad n = rv, \quad k = \frac{n_2}{n_2'} = \frac{r_2}{r_2'}.$

## Generalized class of Rao-regression type estimators under RSS in the presence of non-response

We extend the work of [26, 32] for the estimation of finite population mean when non-response occurs in surveys and RSS is used for the collection of data instead of SRS.

Our first suggested estimator is;

$$t_{r1} = q_1\bar{y}_{rss}^{*} + q_2(\bar{X} - \bar{x}_{rss}^{*}), \tag{20}$$

where $q_1$ and $q_2$ are constants, whose optimum values are used to minimize the error of estimate.

Eq (20) can be written as

$$t_{r1} - \bar{Y} = \bar{Y}(q_1 - 1) + q_1(\bar{y}_{rss}^{*} - \bar{Y}) + q_2(\bar{X} - \bar{x}_{rss}^{*}). \tag{21}$$

The error terms are defined as $\bar{v}_{rss}^{*} = \bar{y}_{rss}^{*} - \bar{Y}$ and $\bar{u}_{rss}^{*} = \bar{X} - \bar{x}_{rss}^{*}$, then it is easy to calculate that;

$$E(\bar{v}_{rss}^{*}) = E(\bar{u}_{rss}^{*}) = 0$$

and

$$E\left(\bar{v}_{rss}^{*2}\right) = V_{yr} = \begin{cases} V'_{yr}, & \text{if RSS is used at second attempt only} \\ V''_{yr}, & \text{if RSS is used at both attempts} \end{cases}$$

$$E\left(\bar{u}_{rss}^{*2}\right) = V_{xr} = \begin{cases} V'_{xr}, & \text{if RSS is used at second attempt only} \\ V''_{xr}, & \text{if RSS is used at both attempts} \end{cases}$$

The bias and MSE of $t_{r1}$, are given by

$$Bias(t_{r1}) \cong \bar{Y}(q_1 - 1).$$

$$MSE(t_{r1}) \cong \bar{Y}^2(q_1 - 1)^2 + q_1^2 \bar{v}_{rss}^{*2} + q_2 \bar{u}_{rss}^{*2} - 2q_1 q_2 \text{Cov}(\bar{v}_{rss}^*, \bar{u}_{rss}^*)$$

or

$$MSE(t_{r1}) \cong \bar{Y}^2(1 + q_1^2 - 2q_1) + q_1^2 V_{yr} + q_2 V_{xr} - 2q_1 q_2 \rho_{yx}\sqrt{V_{yr}V_{xr}}.$$

The optimum values of $q_1$ and $q_2$ with minimum MSE, are given by

$$q_{1opt} = \frac{\bar{Y}^2 V_{xr}}{\bar{Y}^2 V_{xr} + V_{xr}V_{yr} - V_{yxr}^2}, \quad q_{2opt} = \frac{\bar{Y}^2 V_{yxr}}{\bar{Y}^2 V_{xr} + V_{xr}V_{yr} - V_{yxr}^2}$$

and

$$MSE(t_{r1})_{\min} = \frac{\bar{Y}^2(V_{xr}V_{yr} - V_{yxr}^2)}{\bar{Y}^2 V_{xr} + V_{xr}V_{yr} - V_{yxr}^2}. \tag{22}$$

It should be noted that the value of $q_{1opt}$ is mathematically always a positive quantity, while the nature of sign for $q_{2opt}$ is depending upon the correlation coefficient between $Y$ and $X$.

Similarly, our second suggested general class of estimators, is given by

$$t_{ri} = (q_3 \bar{y}_{rss}^* + q_4 \bar{u}_{rss}^*)h(\bar{u}_{rss}^*), \tag{23}$$

where $q_3$ and $q_4$ are pre-determined constants and $h$ is generic function of $\bar{u}_{rss}^* = \bar{X} - \bar{x}_{rss}^*$ which satisfies the following mild conditions;

- Function $h$ is bounded in the vicinity of zeros and is continous.

- Function $h$ is independent of $n$, $N$ and $(X_1, X_2, \ldots, X_N)$.

- Function $h$ is thrice differentiable with bounded and continuous derivatives.

Eq (23) can be written as:

$$t_{ri} = (q_3 \bar{Y} + q_3 \bar{v}_{rss}^* + q_4 \bar{u}_{rss}^*)h(\bar{u}_{rss}^*).$$

Expanding $h(\bar{u}_{rss}^*)$ using Taylor series up to including terms i.e., $O_p(\bar{u}_{rss}^{*2})$, the resulting expression can be written as

$$t_{ri} = q_3 \bar{Y}\left[h(0) + h'(0)\bar{u}_{rss}^* + \frac{1}{2}h''(0)\bar{u}_{rss}^{*2}\right] + q_3 \bar{v}_{rss}^*\left[h(0) + h'(0)\bar{u}_{rss}^*\right]h(\bar{u}_{rss}^*)$$

$$+ q_3 \bar{u}_{rss}^*[h(0) + h'(0)\bar{u}_{rss}^*],$$

where $h(0)$ is a constant term, $h'(0)$ is first order partial derivative in zero and $h''(0)$ is second order partial derivative in zero. For simplicity we write $h(0) = a$, $h'(0) = b$ and $h''(0) = c$. Thus

$$t_{ri} = q_3 \bar{Y} \left[ a + b\bar{u}^*_{rss} + \frac{1}{2} c\bar{u}^{*2}_{rss} \right] + q_3 \bar{v}^*_{rss} [a + b\bar{u}^*_{rss}] h(\bar{u}^*_{rss}) + q_3 \bar{u}^*_{rss} [a + b\bar{u}^*_{rss}]$$

or

$$t_{ri} - \bar{Y} = \bar{Y}(aq_3 - 1) + (aw_1 + bw_0\bar{Y})\bar{u}^*_{rss} + aw_0\bar{v}^*_{rss} + (cw_0\bar{Y} + bw_1)\bar{u}^{*2}_{rss} + bw_0\bar{u}^*_{rss}\bar{v}^*_{rss}. \quad (24)$$

The bias and MSE of $t_{ri}$, are given by

$$Bias(t_{ri}) \cong \bar{Y}(aq_3 - 1) + (cw_0\bar{Y} + bw_1)V_{xr} + bw_0 V_{yxr}.$$

$$\begin{aligned} MSE(t_{ri}) \quad &\cong \bar{Y}^2(aq_1 - 1)^2 + [(aq_2 + bq_1\bar{Y})^2 + 2\bar{Y}(aq_1 - 1)(cq_1\bar{Y} + bq_2)]V_{xr} \\ &+ aq_1^2 V_{yr} + [2aq_1(aq_2 + bq_1\bar{Y}) + 2bq_1\bar{Y}(aq_1 - 1)]\rho_{yx}\sqrt{V_{yr}V_{xr}}. \end{aligned} \quad (25)$$

The optimum values of $q_3$ and $q_4$ are given by

$$q_{3opt} = \frac{\bar{Y}^2 V_{xr}(a^2 - 2b^2 V_{xr} + acV_{xr})}{a^3(\bar{Y}^2 V_{xr} + V_{xr}V_{yr} - \rho_{yx}V_{xr}V_{yr}) + a\bar{Y}^2(2ac - 3b^2)V_{xr}^2}$$

and

$$q_{4opt} = \frac{\bar{Y}[P + b^3\bar{Y}^2 V_{xr}^2 + (a^3\bar{Y} - 2ab^2\bar{Y}V_{xr})\rho_{yx}\sqrt{V_{yr}V_{xr}}]}{V_{xr}[a^4(\bar{Y}^2 + V_{yr} - \rho_{yx}V_{yr}) + a^2\bar{Y}^2(2ac - 3b^2)V_{xr}]}.$$

The minimum MSE of $t_{ri}$, is given as,

$$MSE(t_{ri})_{\min} \cong \frac{\bar{Y}^2[a^4 V_{yr}^3(1 - \rho_{yx}) + b^2\bar{Y}^2 V_{xr}^3(2ac - b^2) - Q]}{a^2[a^2(\bar{Y}^2 + V_{yr} - \rho_{yx}V_{yr}) + \bar{Y}^2 V_{xr}(2ac - 3b^2)]}, \quad (26)$$

where

$$P = a^2 V_{xr}[c\bar{Y}\rho_{yx}V_{yr}V_{yr} - b(\bar{Y}^2 - V_{yr} + \rho_{yx}V_{yr})]$$

and

$$Q = a^2 V_{xr}\{b^2 V_{yr}(1 - \rho_{yx}) + c^2\bar{Y}^2 V_{xr}\}.$$

## Choice of function $h$

The proposed class of estimators is determined by the function which can theoretically and practically take into account a variety of options, but in our study, we discuss the following two well-known functions.

**The ratio function.** We consider [33, 34] estimators as a choice for ratio function $h$.

$$h(\bar{u}^*_{rss}) = \left( \frac{\bar{X}}{\bar{X} - \bar{u}^*_{rss}} \right)^\tau, \quad (27)$$

where $\tau$ is a constant, and

$h(0) = a = 1$,

$h'(0) = b = -\tau,$

$\frac{1}{2} h''(0) = c = \frac{\tau(\tau+1)}{2}.$

**Remarks.** 1. If we consider $\tau = 0$, then $\{a, b, c\} = \{1, 0, 0\}$ and the proposed estimator $t_{ri}$ is converted to proposed estimator $t_{r1}$ as

$$t_{r1} = q_1 \bar{y}^*_{rss} + q_2(\bar{X} - \bar{x}^*_{rss}). \tag{28}$$

2. If we consider $\tau = 1$, then $\{a, b, c\} = \{1, -1, 1\}$ and the function $h$ is converted to Ray and Singh (1981) estimator. Thus the proposed estimator becomes

$$t_{r2} = \left(q_3 \bar{y}^*_{rss} + q_4 \bar{u}^*_{rss}\right)\left(\frac{\bar{X}}{\bar{X} - \bar{u}^*_{rss}}\right). \tag{29}$$

3. If we consider $\tau = -1$, then $\{a, b, c\} = \{1, 1, 0\}$. Hence the proposed estimator is converted to the product type estimator which is used when the correlation coefficient between the study variable and the auxiliary variable is negative i.e.,

$$t_{rp} = \left(q_3 \bar{y}^*_{rss} + q_4 \bar{u}^*_{rss}\right)\left(\frac{\bar{X} - \bar{u}^*_{rss}}{\bar{X}}\right). \tag{30}$$

**The exponential function.** Now we consider the exponential function proposed by [35] as a possible choice of function $h$ i.e.,

$$h\left(\bar{u}^*_{rss}\right) = \exp\left(\frac{\bar{u}^*_{rss}}{2\bar{X} - \bar{u}^*_{rss}}\right) \tag{31}$$

Then proposed class of estimator takes the following form:

$$t_{r3} = \left(q_3 \bar{y}^*_{rss} + q_4 \bar{u}^*_{rss}\right) \exp\left(\frac{\bar{u}^*_{rss}}{2\bar{X} - \bar{u}^*_{rss}}\right). \tag{32}$$

By expanding $h(\bar{u}^*_{rss})$ and using Taylor's series, we get;

$h(0) = a = 1,$

$h'(0) = b = -\frac{1}{2},$

$\frac{1}{2} h''(0) = c = \frac{3}{8}.$

By using the values of constants $a$, $b$ and $c$, we can easily calculate the corresponding optimum values of $q_3$ and $q_4$ with minimum MSE of estimators by using Eq (26), for the two choices of function $h$ discussed above.

## Efficiency comparison

The gain in precision of the proposed class of estimators is completely relying on the precision gained in the estimation by using RSS instead of SRS sampling method. So, it is reasonable to compare the variances of sample mean by using the two competing sampling strategies.

Consider the variance of $\bar{y}_{srs}$

$$Var(\bar{y}_{rss}) = \frac{\sigma_y^2}{n} - \frac{1}{mn}\sum_{i=1}^{m}(\mu_{y(i)} - \mu_y)^2. \tag{33}$$

In the above expression, the term $\sigma_y^2/n$ is same as the variance of $\bar{y}_{srs}$, which indicate that the RSS procedure will produce more precise estimates than the SRS if the inequality $\mu_{y(i)} \neq \mu_y$ holds. This also leads to a certainty that if we use the RSS procedure for collecting data at both attempts, then the proposed class of estimators will be more precise than the estimators discussed under Situation-I, where we collect data through RSS only at second attempt.

## Simulation study

The simulation study is carried out with the same setup used by [28]. We generated the auxiliary variable for two groups as $X_j \sim Normal(N_j, \mu_{yj}, \sigma_{yj}); j = 1, 2$. The corresponding study variable is produced by using the relationship $Y_j = \rho_{yx}X_j + e_j\sqrt{1 - \rho_{yx}^2}$, where $e_j \sim Normal(N_j, 0, 1)$ and $\rho_{yx}$ is the coefficient of correlation between $Y$ and $X$. A sample of size $n = (n_1 + n_2')$ is selected from a population by using the procedures of SRS and RSS and sample means are calculated for the study and the auxiliary variables under Situation-I and Situation-II. Then the sample mean for competing and proposed estimators are estimated. This procedure is repeated 20,000 times to calculate the MSE and RE of the estimators using the following formula;

$$MSE(t_{ri}) = \frac{1}{20,000}\sum_{j=1}^{20,000}(t_{ri} - \mu_y)^2, \tag{34}$$

$$RE(t_{ri}) = \frac{MSE(t_{si})}{MSE(t_{ri})}, \quad i = 1, 2, 3, \tag{35}$$

where $t_{ri}$ represents an estimator under consideration. The results are given in Tables 1–6.

## Conclusions

In this paper, we proposed a generalized class of Rao-regression type estimators for finite population mean under non-response when RSS is used to collect the data at second attempt and at both attempts. Expressions for bias and MSE of the proposed class of estimators were derived up to the first order of approximation. A brief simulation study was conducted to

**Table 1. RE of proposed class of estimators for Situation-I when $k = 2$.**

| $r_2'$ | Estimator | $\rho = 0.10$ | | | $\rho = 0.50$ | | | $\rho = 0.90$ | | |
|---|---|---|---|---|---|---|---|---|---|---|
| | | $v = 3$ | $v = 4$ | $v = 5$ | $v = 3$ | $v = 4$ | $v = 5$ | $v = 3$ | $v = 4$ | $v = 5$ |
| 1 | $t_{r1}$ | 1.510 | 1.520 | 1.563 | 1.502 | 1.516 | 1.530 | 1.591 | 1.603 | 1.620 |
| | $t_{r2}$ | 5.151 | 7.094 | 8.595 | 6.825 | 10.10 | 13.47 | 6.896 | 10.23 | 13.78 |
| | $t_{r3}$ | 3.370 | 4.230 | 4.773 | 6.924 | 10.12 | 13.20 | 7.310 | 10.80 | 14.45 |
| 2 | $t_{r1}$ | 1.500 | 1.514 | 1.562 | 1.580 | 1.580 | 1.610 | 1.570 | 1.583 | 1.680 |
| | $t_{r2}$ | 9.182 | 12.84 | 15.55 | 13.12 | 19.96 | 21.83 | 13.31 | 18.30 | 21.44 |
| | $t_{r3}$ | 5.350 | 6.870 | 8.020 | 13.06 | 19.71 | 21.17 | 13.98 | 19.28 | 23.60 |
| 3 | $t_{r1}$ | 1.572 | 1.573 | 1.540 | 1.610 | 1.701 | 1.720 | 1.690 | 1.721 | 1.740 |
| | $t_{r2}$ | 12.55 | 14.98 | 16.12 | 14.92 | 20.37 | 25.88 | 19.29 | 22.98 | 26.77 |
| | $t_{r3}$ | 6.920 | 9.192 | 10.75 | 14.69 | 21.77 | 27.83 | 15.19 | 22.31 | 27.48 |

**Table 2. RE of proposed class of estimators for Situation-I when $k = 3$.**

| $r_2'$ | Estimator | $\rho = 0.10$ | | | $\rho = 0.50$ | | | $\rho = 0.90$ | | |
|---|---|---|---|---|---|---|---|---|---|---|
| | | $v = 3$ | $v = 4$ | $v = 5$ | $v = 3$ | $v = 4$ | $v = 5$ | $v = 3$ | $v = 4$ | $v = 5$ |
| 1 | $t_{r1}$ | 1.260 | 1.331 | 1.322 | 1.360 | 1.420 | 1.410 | 1.360 | 1.410 | 1.430 |
| | $t_{r2}$ | 5.061 | 7.080 | 7.631 | 5.710 | 8.350 | 11.77 | 5.810 | 9.600 | 12.13 |
| | $t_{r3}$ | 3.273 | 4.120 | 4.051 | 5.760 | 11.19 | 11.40 | 7.250 | 10.19 | 13.91 |
| 2 | $t_{r1}$ | 1.341 | 1.321 | 1.360 | 1.390 | 1.400 | 1.460 | 1.400 | 1.420 | 1.460 |
| | $t_{r2}$ | 8.962 | 11.92 | 14.65 | 12.50 | 16.32 | 21.18 | 12.79 | 14.70 | 16.86 |
| | $t_{r3}$ | 5.053 | 6.030 | 7.460 | 10.36 | 15.99 | 20.38 | 11.49 | 16.78 | 21.26 |
| 3 | $t_{r1}$ | 1.350 | 1.351 | 1.370 | 1.450 | 1.440 | 1.520 | 1.510 | 1.540 | 1.520 |
| | $t_{r2}$ | 11.63 | 12.99 | 14.74 | 12.18 | 16.60 | 21.65 | 17.48 | 20.22 | 21.29 |
| | $t_{r3}$ | 6.601 | 9.140 | 10.04 | 12.06 | 20.99 | 24.81 | 13.55 | 21.70 | 22.42 |

**Table 3. RE of proposed class of estimators for Situation-I when $k = 4$.**

| $r_2'$ | Estimator | $\rho = 0.10$ | | | $\rho = 0.50$ | | | $\rho = 0.90$ | | |
|---|---|---|---|---|---|---|---|---|---|---|
| | | $v = 3$ | $v = 4$ | $v = 5$ | $v = 3$ | $v = 4$ | $v = 5$ | $v = 3$ | $v = 4$ | $v = 5$ |
| 1 | $t_{r1}$ | 1.103 | 1.101 | 1.133 | 1.113 | 1.140 | 1.172 | 1.124 | 1.172 | 1.195 |
| | $t_{r2}$ | 4.781 | 6.951 | 6.958 | 5.387 | 9.241 | 9.670 | 5.520 | 9.522 | 12.12 |
| | $t_{r3}$ | 3.240 | 4.103 | 4.031 | 4.358 | 9.960 | 9.102 | 6.974 | 11.17 | 12.93 |
| 2 | $t_{r1}$ | 1.153 | 1.161 | 1.160 | 1.183 | 1.192 | 1.210 | 1.180 | 1.195 | 1.251 |
| | $t_{r2}$ | 8.152 | 10.08 | 12.56 | 11.18 | 15.98 | 16.25 | 10.47 | 11.58 | 13.08 |
| | $t_{r3}$ | 4.311 | 5.682 | 6.831 | 9.970 | 12.22 | 14.98 | 10.24 | 14.73 | 18.69 |
| 3 | $t_{r2}$ | 1.201 | 1.212 | 1.242 | 1.242 | 1.251 | 1.302 | 1.274 | 1.345 | 1.412 |
| | $t_{r2}$ | 10.77 | 11.08 | 13.04 | 11.43 | 13.36 | 15.04 | 15.78 | 18.03 | 20.38 |
| | $t_{r3}$ | 5.123 | 8.570 | 9.621 | 11.10 | 16.50 | 22.73 | 16.99 | 17.70 | 22.72 |

**Table 4. RE of proposed class of estimators for Situation-II when $k = 2$.**

| $r_1, r_2'$ | Estimator | $\rho = 0.10$ | | | $\rho = 0.50$ | | | $\rho = 0.90$ | | |
|---|---|---|---|---|---|---|---|---|---|---|
| | | $v = 3$ | $v = 4$ | $v = 5$ | $v = 3$ | $v = 4$ | $v = 5$ | $v = 3$ | $v = 4$ | $v = 5$ |
| (3, 1) | $t_{r1}$ | 1.932 | 1.856 | 1.940 | 2.022 | 2.011 | 2.088 | 2.003 | 1.992 | 2.040 |
| | $t_{r2}$ | 5.487 | 6.353 | 7.450 | 8.332 | 11.18 | 14.50 | 8.470 | 11.43 | 15.11 |
| | $t_{r3}$ | 3.549 | 3.760 | 4.232 | 8.051 | 10.57 | 13.27 | 8.732 | 11.75 | 15.42 |
| (3, 2) | $t_{r1}$ | 1.809 | 1.830 | 1.670 | 2.010 | 2.006 | 1.962 | 1.990 | 1.950 | 1.920 |
| | $t_{r2}$ | 7.152 | 8.882 | 12.01 | 11.74 | 16.56 | 22.35 | 11.96 | 17.04 | 23.15 |
| | $t_{r3}$ | 4.185 | 4.846 | 5.762 | 11.22 | 15.39 | 20.28 | 12.28 | 17.39 | 23.54 |
| (3, 3) | $t_{r1}$ | 1.760 | 1.605 | 1.650 | 1.982 | 1.935 | 2.007 | 1.940 | 1.903 | 1.930 |
| | $t_{r2}$ | 8.772 | 10.26 | 12.34 | 14.58 | 22.14 | 29.18 | 14.88 | 22.72 | 30.27 |
| | $t_{r3}$ | 4.880 | 5.122 | 5.878 | 13.85 | 20.46 | 26.39 | 15.22 | 23.14 | 30.63 |

**Table 5. RE of proposed class of estimators for Situation-II when $k = 3$.**

| $r_1, r'_2$ | Estimator | $\rho = 0.10$ | | | $\rho = 0.50$ | | | $\rho = 0.90$ | | |
|---|---|---|---|---|---|---|---|---|---|---|
| | | $v = 3$ | $v = 4$ | $v = 5$ | $v = 3$ | $v = 4$ | $v = 5$ | $v = 3$ | $v = 4$ | $v = 5$ |
| (3, 1) | $t_{r1}$ | 1.773 | 1.844 | 1.550 | 1.940 | 2.184 | 1.794 | 1.930 | 2.186 | 1.790 |
| | $t_{r2}$ | 4.511 | 5.926 | 5.584 | 6.805 | 10.99 | 11.16 | 6.911 | 11.33 | 11.49 |
| | $t_{r3}$ | 2.970 | 3.499 | 3.168 | 6.528 | 10.17 | 10.23 | 7.036 | 11.47 | 11.65 |
| (3, 2) | $t_{r1}$ | 1.613 | 1.548 | 1.430 | 1.974 | 1.993 | 1.930 | 1.980 | 2.007 | 1.950 |
| | $t_{r2}$ | 5.423 | 6.466 | 6.900 | 8.900 | 12.34 | 15.58 | 9.082 | 12.56 | 15.92 |
| | $t_{r3}$ | 3.270 | 3.566 | 3.563 | 8.468 | 11.58 | 14.31 | 9.220 | 12.75 | 16.10 |
| (3, 3) | $t_{r1}$ | 1.493 | 1.406 | 1.333 | 1.914 | 1.920 | 1.840 | 1.916 | 1.940 | 1.868 |
| | $t_{r2}$ | 6.080 | 7.110 | 7.610 | 10.49 | 14.54 | 18.85 | 10.63 | 14.67 | 19.00 |
| | $t_{r3}$ | 3.463 | 3.673 | 3.734 | 9.996 | 13.68 | 17.46 | 10.80 | 14.88 | 19.24 |

**Table 6. RE of proposed class of estimators for Situation-II when $k = 4$.**

| $r_1, r'_2$ | Estimator | $\rho = 0.10$ | | | $\rho = 0.50$ | | | $\rho = 0.90$ | | |
|---|---|---|---|---|---|---|---|---|---|---|
| | | $v = 3$ | $v = 4$ | $v = 5$ | $v = 3$ | $v = 4$ | $v = 5$ | $v = 3$ | $v = 4$ | $v = 5$ |
| (3, 1) | $t_{r1}$ | 1.690 | 1.547 | 1.535 | 1.995 | 1.902 | 1.949 | 2.009 | 1.935 | 1.970 |
| | $t_{r2}$ | 4.458 | 4.748 | 4.990 | 7.738 | 9.548 | 11.87 | 7.838 | 9.630 | 12.18 |
| | $t_{r3}$ | 2.833 | 2.812 | 2.864 | 7.365 | 8.977 | 10.82 | 7.942 | 9.794 | 12.30 |
| (3, 2) | $t_{r1}$ | 1.479 | 1.349 | 1.282 | 1.934 | 1.870 | 1.790 | 1.980 | 1.954 | 1.872 |
| | $t_{r2}$ | 4.461 | 4.970 | 5.070 | 8.536 | 11.96 | 14.72 | 8.657 | 12.00 | 14.88 |
| | $t_{r3}$ | 2.733 | 2.765 | 2.742 | 8.095 | 11.22 | 13.46 | 8.762 | 12.20 | 15.07 |
| (3, 3) | $t_{r1}$ | 1.360 | 1.213 | 1.219 | 1.932 | 1.706 | 1.793 | 2.004 | 1.797 | 1.901 |
| | $t_{r2}$ | 4.739 | 4.961 | 5.168 | 9.860 | 14.03 | 17.38 | 9.887 | 14.05 | 17.35 |
| | $t_{r3}$ | 2.755 | 2.670 | 2.730 | 9.447 | 13.03 | 15.98 | 10.03 | 14.27 | 17.61 |

Simulation results in Tables 1–6 show that the RE of the proposed class of estimators under RSS is higher than its competing estimators under SRS for estimating population mean. It is also observed that the RE increases when the correlation coefficient between $Y$ and $X$ is increased. The RE also increases by increasing the overall sample size $n$, while it decreases when the value of non-response rate $k$ is increased. We also observed that the RE is comparatively higher for a moderate set size $v$, i.e., $v = 4$. Furthermore, the RE is higher for Situation-I when compared to Situation-II of sampling selection under RSS procedure for the same sample size.

determine the RE of proposed class of estimators for different values of non-response rate $k$, set size $v$, and overall sample size $n$.

Based on simulation results, we suggest using the proposed class of estimators under RSS for more precise estimation of the finite population mean when non-response occurs in the data. Additionally, we encourage using the RSS technique for data collection in both attempts since it provides more precise estimates.

## Supporting information

**S1 Appendix.**
(PDF)

## Acknowledgments

I would like to express gratitude to my PhD supervisor, Prof. Dr. Javid Shabbir, who helped me write this article.

## Author Contributions

**Conceptualization:** Syed Abdul Rehman.

**Data curation:** Syed Abdul Rehman.

**Formal analysis:** Syed Abdul Rehman.

**Methodology:** Syed Abdul Rehman.

**Supervision:** Javid Shabbir.

**Writing – original draft:** Syed Abdul Rehman.

**Writing – review & editing:** Javid Shabbir.

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
