## [Editor Report · Decision Letter 0]

30 Jun 2022

PONE-D-22-15506An efficient class of estimators for the population mean in the presence of non-response under Ranked Set SamplingPLOS ONE

Dear Dr. Syed Abdul Rehman,

Thank you for submitting your manuscript to PLOS ONE. After careful consideration, we feel that it has merit but does not fully meet PLOS ONE’s publication criteria as it currently stands. Therefore, we invite you to submit a revised version of the manuscript that addresses the points raised during the review process. The paper is poorly written and cannot be sent to reviewers before serious rewriting and editing. I stocked at the very beginning of Section 1.1. Here are some of my preliminary comments and questions:

1. As follows from the first sentence, the authors claim that estimation of population parameters are unreliable if the distribution of the general population is asymmetric. This is a false statement. Many distributions we deal with are asymmetric, like exponential, gamma, etc. I’ve never heard that parameters estimators are unreliable. What the authors mean?

2. The second sentence is about response and non-response. Do the authors talk about surveys? What is “pattern?” Do they mean that people who do not response constitute a different distribution? Can’t we treat them as missing observations?

3. English is bad in the third sentence of Section 1.1: “This appeals that in the existence…” I simply do not understand what the authors wanted to say.

4. The problem set up in Section 2 makes little sense. How can we divide the general population Ω into response group Ω1 and non-response group Ω2 before surveys have been conducted? We can split a sample into response and non-response groups not a general population. The language is so sloppy that the reader even does not understand the problem set up.

 Please submit your revised manuscript by Aug 14 2022 11:59PM. If you will need more time than this to complete your revisions, please reply to this message or contact the journal office at plosone@plos.org. Please include the following items when submitting your revised manuscript:A rebuttal letter that responds to each point raised by the academic editor and reviewer(s). You should upload this letter as a separate file labeled 'Response to Reviewers'.A marked-up copy of your manuscript that highlights changes made to the original version. You should upload this as a separate file labeled 'Revised Manuscript with Track Changes'.An unmarked version of your revised paper without tracked changes. You should upload this as a separate file labeled 'Manuscript'.

We look forward to receiving your revised manuscript.

Kind regards,

Eugene Demidenko, Ph.D.

Academic Editor

PLOS ONE

Journal Requirements:

"This study is made without any funding agency. This study is part of my PhD thesis for which synopsis is approved by BSR at quaid i azam university, Islamabad, Pakistan."

"NO, authors have no competing interests."

---

## [Author Response · Author response to Decision Letter 0]

14 Aug 2022

The link of "View Decision Letter" is blank and blocked, however Response to the reviewer file is uploaded.

---

## [Decision Letter · Decision Letter 1]

15 Sep 2022

PONE-D-22-15506R1An efficient class of estimators for finite population mean in the presence of non-response under ranked set sampling (RSS)PLOS ONE

Dear Dr. Syed Abdul Rehman,

Thank you for submitting your manuscript to PLOS ONE. After careful consideration, we feel that it has merit but does not fully meet PLOS ONE’s publication criteria as it currently stands. Therefore, we invite you to submit a revised version of the manuscript that addresses the points raised during the review process.

Please carefully address all reviewers comments. Especially important are issues related to English. Your paper must be read and edited by an English proficient expert. Failure to omit response to any point of critique may result in rejection of the paper.

We look forward to receiving your revised manuscript.

Kind regards,

Eugene Demidenko, Ph.D.

Academic Editor

PLOS ONE

Reviewers' comments:

Reviewer's Responses to Questions

**Comments to the Author**

1. If the authors have adequately addressed your comments raised in a previous round of review and you feel that this manuscript is now acceptable for publication, you may indicate that here to bypass the “Comments to the Author” section, enter your conflict of interest statement in the “Confidential to Editor” section, and submit your "Accept" recommendation.

Reviewer #1: (No Response)

Reviewer #2: (No Response)

Reviewer #3: (No Response)

Reviewer #4: All comments have been addressed

Reviewer #5: (No Response)

2. Is the manuscript technically sound, and do the data support the conclusions?

Reviewer #1: Yes

Reviewer #2: Yes

Reviewer #3: Yes

Reviewer #4: Yes

Reviewer #5: Yes

3. Has the statistical analysis been performed appropriately and rigorously? 

Reviewer #1: Yes

Reviewer #2: Yes

Reviewer #3: Yes

Reviewer #4: Yes

Reviewer #5: (No Response)

4. Have the authors made all data underlying the findings in their manuscript fully available?

Reviewer #1: Yes

Reviewer #2: Yes

Reviewer #3: Yes

Reviewer #4: Yes

Reviewer #5: (No Response)

5. Is the manuscript presented in an intelligible fashion and written in standard English?

Reviewer #1: Yes

Reviewer #2: Yes

Reviewer #3: Yes

Reviewer #4: Yes

Reviewer #5: (No Response)

6. Review Comments to the Author

Reviewer #1: 1. Authors need to update the introduction by incorporating the latest references regarding L-moments use.

2. Extend results interpretation.

3. Conclusion should be precise.

Reviewer #2: Estimation of unknown mean under RSS frame work and in situation of non response is discussed. Some estimators are proposed and their properties are studied. A simulation study is carried out in support of theoretical study.

Line 206- estimators is provides...some other language issues are there, read whole manuscript minutely

Reviewer #3: 1) In the simulation study, I am not if results remain unchanged for negative values of rho, e.g. for rho=+/-0.5. Could authors comment on this? If the answer is negative, the negative values can be considered in the study as well.

2) The simulation results are described very briefly. It definitely needs more elaboration.

3) When an acronym (like RSS) is introduced, then it should be used instead of the full form. This simple rule has been violated in the text.

4) A paragraph should be added to the beginning of Conclusion that outlines what has been done.

5) There is some room to improve English writing. Some specific corrections are highlighted in the attached pdf file (I strongly recommend authors to use language editing service offered by academic centers). Also, there are some problems in the Reference section. Please carefully fix them.

Reviewer #4: Paper ID: PONE-D-22-15506R1

Report of the paper “An efficient class of estimators for finite population mean in the presence of non-response under ranked set sampling (RSS)” Submitted to PLOS ONE

Referee’s Comments

In this paper, authors have suggested a class of Rao-regression type estimators and ranked set sampling (RSS) scheme is used to collect data from different situations non-response response

Expressions for bias and mean square error of the estimators are obtained up to first order of approximation and comprehensive simulation study is carried out to observe the performances of the proposed estimators under non-response. Authors have incorporated all the comments raised by the previous reviewer and I agreed with the response. The idea/concept of the paper is interesting and can be published after some corrections. Some suggestions are as follows:

1) I would like to suggest the authors to make an Appendix to post all codes related to this paper in the next submission.

2) Add these references in the reference section as well in the text section

a. Khalid (2019). Effective Estimation Strategy of Population Variance in Two-Phase Successive Sampling Under Random Non-response. J Stat Theory Pract 13, 4 https://doi.org/10.1007/s42519-018-0010-y

b. Mohd Khalid. "Exponential chain dual to ratio and regression type estimators of population mean in two-phase sampling." Statistica 75.4 (2015): 379-389.

c. Khalid, Mohd. "Some imputation methods to deal with the issue of missing data problems due to random non-response in two-occasion successive sampling." Communications in Statistics-Simulation and Computation (2020): 1-21.

Reviewer #5: Review report on “An efficient class of estimators for finite population mean in the presence of non-response under ranked set sampling (RSS)”

The manuscript deals with the problem of estimation of population mean in presence of non-response on study character. The properties of the suggested generalized class of Rao-regression type estimators under RSS in presence of non-response have been studied and efficiency has been compared through simulation study.

Although the problem considered in the manuscript has been found to be interesting and mathematically robust but there is some suspicion of incorrectness or typographical error in some of the expressions, and so minor revision is required.

1. On page 5, the expression shown in equations (13), (18) and (19) appears to be erroneous, it should be verified properly.

2. On page 6, "RSS only in second attempt and both attempts" the situation should be corrected.

7. PLOS authors have the option to publish the peer review history of their article (what does this mean?). If published, this will include your full peer review and any attached files.

Reviewer #1: No

Reviewer #2: No

Reviewer #3: No

Reviewer #4: No

Reviewer #5: No

---

## [Author Response · Author response to Decision Letter 1]

18 Oct 2022

Both authors are thankful to the editor and all the reviewers for their valueable time to review our manuscript. All the suggestions and corrections pointed out by the reviewers are considered and manuscript is revised and corrected carefully. Response to each reviewer is written and uploaded as "Response to reviewers" on this website.

---

## [Editor Report · Decision Letter 2]

24 Oct 2022

An efficient class of estimators for finite population mean in the presence of non-response under ranked set sampling (RSS)

PONE-D-22-15506R2

Dear Dr. Syed Abdul Rehman,

We’re pleased to inform you that your manuscript has been judged scientifically suitable for publication and will be formally accepted for publication once it meets all outstanding technical requirements.

Kind regards,

Eugene Demidenko, Ph.D.

Academic Editor

PLOS ONE
---

## [Editor Report · Acceptance letter]

28 Oct 2022

PONE-D-22-15506R2 

An efficient class of estimators for finite population mean in the presence of non-response under ranked set sampling (RSS) 

Dear Dr. Rehman:

I'm pleased to inform you that your manuscript has been deemed suitable for publication in PLOS ONE. Congratulations! Your manuscript is now with our production department. 

Kind regards, 

on behalf of

Dr. Eugene Demidenko 

Academic Editor

PLOS ONE